# Root Metabolism and Effects of Root Exudates on the Growth of *Ralstonia solanacearum* and *Fusarium moniliforme* Were Significantly Different between the Two Genotypes of Peanuts

**DOI:** 10.3390/genes14020528

**Published:** 2023-02-20

**Authors:** Zhong Li, Wenfeng Guo, Changming Mo, Ronghua Tang, Liangqiong He, Lin Du, Ming Li, Haining Wu, Xiumei Tang, Zhipeng Huang, Xingjian Wu

**Affiliations:** 1Guangxi Crop Genetic Improvement and Biotechnology Laboratory, Guangxi Academy of Agricultural Sciences, Nanning 530004, China; 2Cash Crops Research Institute, Guangxi Academy of Agricultural Sciences, Nanning 530004, China; 3Guangxi Science and Technology Museum, Nanning 530016, China

**Keywords:** peanut, root exudates, differentially expressed genes, differentially expressed metabolites, pathogens, interaction

## Abstract

Wild peanut species *Arachis correntina* (*A. correntina*) had a higher continuous cropping tolerance than peanut cultivars, closely correlating with the regulatory effects of its root exudates on soil microorganisms. To reveal the resistance mechanism of *A. correntina* to pathogens, we adopted transcriptomic and metabolomics approaches to analyze differentially expressed genes (DEGs) and differentially expressed metabolites (DEMs) between *A. correntina* and peanut cultivar Guihua85 (GH85) under hydroponic conditions. Interaction experiments of peanut root exudates with *Ralstonia solanacearum* (*R. solanacearum*) and *Fusarium moniliforme* (*F. moniliforme*) were carried out in this study. The result of transcriptome and metabolomics association analysis showed that there were fewer up-regulated DEGs and DEMs in *A. correntina* compared with GH85, which were closely associated with the metabolism of amino acids and phenolic acids. Root exudates of GH85 had stronger effects on promoting the growth of *R. solanacearum* and *F. moniliforme* than those of *A. correntina* under 1 and 5 percent volume (1% and 5%) of root exudates treatments. Thirty percent volume (30%) of *A. correntina* and GH85 root exudates significantly inhibited the growth of two pathogens. The exogenous amino acids and phenolic acids influenced *R. solanacearum* and *F. moniliforme* showing concentration effects from growth promotion to inhibition as with the root exudates. In conclusion, the greater resilience of *A. correntina*) to changes in metabolic pathways for amino acids and phenolic acids might aid in the repression of pathogenic bacteria and fungi.

## 1. Introduction

Continuous cropping has a significant negative impact on crop productivity in part due to the buildup of soil pathogens [1]. The deterioration of rhizosphere soil microflora under continuous cropping systems reduces the amount of soil bacteria and actinomycetes while increasing the amount of soil fungi [1,2,3], correlated with enhanced pathogenicity through an imbalance in the soil microbiome [4]. Cultural methods and chemical applications can alleviate problems but are costly and bring environmental concerns [5,6]. The application of improved varieties is considered a promising alternative treatment that circumvents the above problems.

Most peanut cultivars are intolerant to continuous cropping in part due to bacterial wilt and root rot, caused by *R. solanacearum* and *F. moniliforme*, respectively [7,8]. The control of these two diseases is of great significance to the sustainable development of the peanut industry. The resistance of wild peanut species to certain diseases and insect pests is significantly higher than that of peanut cultivars, such as higher resistance to thrips, spot and rust pathogens [9]. According to the classification of Krapovickas, *A. correntina* belongs to the peanut perennial flora [10]. Moreover, *A. correntina* is one of the few wild peanut species that has a better cross-compatibility in distant hybridizations, which is beneficial to its application in the breeding of peanut cultivars [11]. *A. correntina* shows stronger physiological and ecological adaptability under continuous cropping conditions than peanut cultivars. This may be caused by its difference from peanut cultivars in root secretion, rhizosphere microflora and microbial activities, soil nutrient uptake and utilization [11]. GH85 is a high-yield and susceptible peanut cultivar relative to *A. correntina* that allows for comparative tests.

The interaction between crop diseases and root exudates is a key point in the study of continuous cropping. Plants actively secrete up to 40% of photosynthetically fixed carbon into the rhizosphere soil that promotes or inhibits the growth of specific microorganisms, leading to multiple types of physical and chemical interactions between microorganisms and plants. These interactions can be neutral, beneficial or harmful when plant-pathogenic microorganisms are involved [12,13,14]. A previous study indicates that rhizosphere soils of highly susceptible plants may have more diverse and/or more abundant microbial communities than those of resistant plants [15]. However, the role of root exudates in the pathogenesis of root-infected bacteria and fungi has not been fully understood, partly due to the inadequacy of available analytical methods [16,17,18].

This paper explores the role of the root exudates in the differential sensitivity to pathogens between wild peanut species and peanut cultivars. Transcriptomic and metabolite analyses were employed. The differences in DEGs and DEMS were characterized along with understanding how the root exudates influenced pathogen growth.

## 2. Materials and Methods

### 2.1. Plant Materials and Treatments

Wild peanut wild species *A. correntina* and cultivated variety GH85 were used for the experiment. *A. correntina* was obtained from the National Wild Peanut Germplasm Nursery (Nanning, Guangxi, China). GH85 was provided by Guangxi Academy of Agricultural Sciences Economic Crops Research Institute (Nanning, Guangxi, China). *R. solanacearum* (Strain Number: ACCC 60145) and *F. moniliforme* (Strain Number: ACCC 36127) were obtained from Agricultural Culture Collection of China (Beijing, China), isolated from infected peanut tissue. The experiment was carried out on 18 May 2021 in Guangxi Crop Genetic Improvement and Biotechnology Laboratory (Nanning, Guangxi, China).

### 2.2. Collection of Peanut Root Exudates

The seeds of *A. correntina* and GH85 were wetted and cultured in a petri dish at 28 °C for 48 h. Then, eight peanut germinated seeds were sown in a 2 L glass culture cup (height 20 cm, width 13.3 cm) containing 1.6 L 1/4 Murashige and Skoog (MS) liquid medium (MS powder 0.55 g/L) [19] with plastic plates at 26 °C and light incubator cultured for 30 d and 60 d, with no peanut seeds as the control; each treatment was repeated three times. The MS liquid medium was adjusted to pH 5.9 with KOH_2_. The hydroponic liquid was collected and replaced every 15 days. The hydroponic solutions were combined as root exudatesⅠ 0–30 days after sowing. Hydroponic solutions were combined as root exudatesⅡ 30–60 days after sowing. The collected root exudatesⅠ and root exudatesⅡ were freeze-dried at 4 °C, respectively, and at last, the total volume of root exudate concentrated solutions was 300 mL, stored at −20 °C for future use.

### 2.3. Detection and Analysis of Peanut Root Exudates 

One ml concentrated solutions of root exudatesⅠ, root exudatesⅡ and control were extracted overnight at 4 °C with 1.2 mL 70% aqueous methanol, each treatment repeated three times. Following centrifugation at 10,000× *g* for 10 min, the extracts were absorbed (CNWBOND Carbon-GCB SPE Cartridge, 250 mg, 3 mL; ANPEL, Shanghai, China, www.anpel.com.cn/cnw, accessed on 12 August 2021) and filtrated (SCAA-104, 0.22 μm pore size; ANPEL, Shanghai, China, http://www.anpel.com.cn/, accessed on 12 August 2021) before UPLC-MS/MS analysis.

The sample extracts were analyzed using a UPLC-ESI-MS/MS system (UPLC, Shim-pack UFLC SHIMADZU CBM30A system, www.shimadzu.com.cn/, accessed on 12 August 2021; MS, Applied Biosystems 6500 Q TRAP, www.appliedbiosystems.com.cn/, accessed on 12 August 2021). The analytical conditions were as follows, UPLC: column, Waters ACQUITY UPLC HSS T3 C18 (1.8 µm, 2.1 mm × 100 mm); the mobile phase consisted of solvent A, pure water with 0.04% acetic acid, solvent B and acetonitrile with 0.04% acetic acid. Sample measurements were performed with a gradient program that employed the starting conditions of 95% A and 5% B. Within 10 min, a linear gradient to 5% A and 95% B was programmed, and a composition of 5% A and 95% B was kept for 1min. Subsequently, a composition of 95% A and 5% B was adjusted within 0.10 min and kept for 2.9 min. The column oven was set to 40 °C; the injection volume was 2 μL. The effluent was alternatively connected to an ESI-triple quadrupole-linear ion trap (Q TRAP)-MS.

LIT and triple quadrupole (QQQ) scans were acquired on a triple quadrupole-linear ion trap mass spectrometer (Q TRAP), API 6500 Q TRAP UPLC/MS/MS System, equipped with an ESI Turbo Ion-Spray interface, operating in positive and negative ion mode and controlled by Analyst 1.6.3 software (AB Sciex, Shang hai, China). The ESI source operation parameters were as follows: ion source, turbo spray; source temperature 550 °C; ion spray voltage (IS) 5500 V (positive ion mode)/−4500 V (negative ion mode); ion source gas I (GSI), gas II(GSII) and curtain gas (CUR) were set at 50, 60 and 30.0 psi, respectively. The collision gas(CAD) was high. Instrument tuning and mass calibration were performed with 10 and 100 μmol/L polypropylene glycol solutions in QQQ and LIT modes, respectively [20]. QQQ scans were acquired as MRM experiments with collision gas (nitrogen) set to 5 psi. DP and CE for individual MRM transitions were performed with further DP and CE optimization [21]. A specific set of MRM transitions were monitored for each period according to the metabolites eluted within this period.

Quality control (QC) samples were prepared by blending the extracts of samples in all groups with equal amounts. Four replicate QC samples were prepared separately, named mix01 to mix04. QC samples were determined with the same method as well as the samples. In instrumental analysis, one QC sample was determined after every three samples to evaluate the repeatability of the whole analysis process. Significantly different metabolites between groups were determined by VIP ≥ 1 and absolute log2FC (fold change) ≥ 1. Unsupervised principal component analysis (PCA) was performed by the Past software (https://www.macupdate.com/app/mac/62317/past, accessed on 14 August 2021). The stratified clustering of samples by heat map was performed using the EBSeqR package [22]. Identified metabolites were annotated using the KEGG compound database, and annotated metabolites were then mapped to the KEGG pathway database. Pathways mapped with significantly regulated metabolites were then utilized for MSEA (metabolite set enrichment analysis), and their significance was determined by *p*-values of hypergeometric tests [23].

### 2.4. RNA Extraction, Library Construction, RNA-Sequencing and Data Analysis

Root RNA was isolated from peanut root tips on the 30th day and the 60th day after hydroponics, respectively. RNA was extracted from the roots of each sample using a total RNA Purification Kit (TIANDZ, Beijing, China) treated with DNase I (TaKaRa Biotechnology, Dalian, China); each treatment was repeated three times. RNA concentration was measured using NanoDrop 2000 (Thermo, Wuhan, China). RNA integrity was assessed using the RNA Nano 6000 Assay Kit of the Agilent Bioanalyzer 2100 system (Agilent Technologies, Wuhan, China, Los Angeles, CA, USA). A total amount of 1 μg RNA per sample was used as input material for the RNA sample preparations. Sequencing libraries were generated using NEBNextUltra TM RNA Library Prep Kit for Illumina (NEB, Wuhan, China, Los Angeles, CA, USA) following manufacturer’s recommendations, and index codes were added to attribute sequences to each sample. Briefly, mRNA was purified from total RNA using poly-T oligo-attached magnetic beads. Fragmentation was carried out using divalent cations under elevated temperature in NEBNext First Strand Synthesis Reaction Buffer (5X). First strand cDNA was synthesized using random hexamer primer and M-MuLV Reverse Transcriptase (RNase H-). Second strand cDNA synthesis was subsequently performed using DNA Polymerase I and RNase H. Remaining overhangs were converted into blunt ends via exonuclease/polymerase activities. After adenylation of 3′ ends of DNA fragments, NEBNext Adaptor with hairpin loop structure was ligated to prepare for hybridization. In order to select cDNA fragments of preferentially 240 bp in length, the library fragments were purified with AMPure XP system (Beckman Coulter, Beverly, MA, USA). Then, 3 μL USER Enzyme (NEB, USA) was used with size-selected, adaptor-ligated cDNA at 37 °C for 15 min followed by 5 min at 95 °C before PCR. Then, PCR was performed with Phusion High-Fidelity DNA polymerase, Universal PCR primers and Index (X) Primer. At last, PCR products were purified (AMPure XP system) and library quality was assessed on the Agilent Bioanalyzer 2100 system.

Prior to DEG analysis, raw reads were edited to filter out low-quality reads containing adaptor sequences in order to obtain clean reads. These clean reads were then mapped to the reference genome sequence. Only reads with a perfect match or one mismatch were further analyzed and annotated based on the reference genome by using Hisat2 tools soft. The reference genome was derived from the peanut base (Tifrunner.gnm2.ann1.4KOL). DEG analysis of two samples was performed using the EBSeqR package [22]. The resulting FDR (false discovery rate) was adjusted using the PPDE (posterior probability of being DE). The FDR < 0.05 and|log_2_ (foldchange)| ≥ 1 was set as the threshold for significant differential expression. Gene function was annotated based on the following databases: Nr (NCBI non-redundant protein sequences), Nt (NCBI non-redundant nucleotide sequences), Pfam (Protein family), KOG/COG (Clusters of Orthologous Groups of proteins), Swiss-Prot (A manually annotated and reviewed protein sequence database), KO (KEGG Ortholog database) and GO (Gene Ontology).

### 2.5. qRT-PCR Analyses

Two µg of total root RNA isolated as described above was reverse transcribed using M-MLV reverse transcriptase (Invitrogen, Wuhan, China, Los Angeles, CA, USA). Quantitative real-time PCR (qRT-PCR) was performed with three biological and technical replicates by using SYBR^®^ Premix EX Taq™ (Takara, Tokyo, Japan). Primers were designed by using Premier 3.0 primer design software online for qRT-PCR. The PCR amplification conditions consisted of one cycle of 95 °C for 30 s, followed by 40 cycles of 95 °C for 5 s and 60 °C for 20 s. The formula 2^−ΔΔCT^ method was employed to calculate relative expression level of candidate genes [24].

### 2.6. Interaction between Two Pathogens and Peanut Root Exudates 

Root exudatesⅠ, root exudatesⅡ and 12 compounds detected from peanut root exudates were used in interaction tests. Twelve compounds included 7 phenolic acid [3-Aminosalicylic Acid, 2,4-Dihydroxybenzoic Acid, 3-(4-Hydroxyphenyl)-propionic Acid, Syringic Aldehyde, 2-Methoxybenzoic acid, p-Coumaric Acid, Ferulic Acid] and 5 amino acids (Tryptophan, L-Proline, L-Valine, L-Methionine, L-Aspartic Acid). Three treatments were set for the interaction tests of root exudates and *F. moniliforme* by root exudatesⅠ and root exudatesⅡ mixed with martin medium (Peptone 5.0 g/L, Dextrose 10.0 g/L, Monopotassium phosphate 1.0 g/L, Magnesium Sulfate 0.5 g/L, Chloramphenicol 0.1 g/L, Rose Bengal 0.033 g/L, Agar 18.0 g/L) in a ratio of 1:99 (1 volume percent), 5:95 (5 volume percent) and 30:79 (30 volume percent), respectively. Twelve compounds were mixed with martin medium to prepare three concentration treatments of 0.001, 0.01 and 0.1 mol/L for the interaction tests. The controls were treated in the same way using deionized water. After full blending, the mixture medium was transferred to a 12 cm diameter petri dish. Mycelium growth rate method was used to determine the antibacterial activity of each root exudate treatment, and the 12 compounds against *F. moniliforme*. *F. moniliforme* cakes were prepared by using a hole punch with an inner diameter of 5 mm on the edge of the precultured *F. moniliforme* colony and then inoculated on the treatments and control, respectively. Each treatment was repeated three times. After incubating for 3d and 4d in a constant temperature incubator at 28 °C, the colony diameters of *F. moniliforme* were measured by cross method and the mean values were taken. The inhibition rate of different concentrations of root exudates and 12 compounds on the growth of *F. moniliforme* mycelia was analyzed. Inhibition rate of *F. moniliforme* colony growth = [(colony circle diameter of treatment group − colony circle diameter of control group)/colony circle diameter of control group] × 100%.

Root exudatesⅠ and root exudatesⅡ were mixed with PDA medium (potato extract 4.0 g/L, dextrose 20.0 g/L, Agar 15.0 g/L) in a ratio of 1:99 (1 volume percent), 5:95 (5 volume percent) and 30:79 (30 volume percent), respectively, to prepare three treatments for the interaction tests of root exudates and *R. solanacearum*. Twelve compounds were mixed with PDA medium to prepare three concentration treatments of 0.001, 0.01 and 0.1 mol/L for the interaction tests. The controls were treated in the same way using deionized water. Each treatment was repeated three times. Then, 1.5 mL of above mediums mixed with 20 μL dilute liquid of *R. solanacearum* (1 × 10^−6^ cfu/mL) (cfu, colony forming unit) cultured for 6 h and 200 μL of above mixed culture medium was coated on a 12 cm diameter petri dish containing PDA solid medium, and each was treated with three dishes, cultured in a constant temperature incubator at 28 °C for 24 h. The bacterial colony and bacterial inhibition rate were counted. Inhibition rate of *R. solanacearum* (%) = (number of treatment colony − number of control colony)/number of control colony × 100%.

## 3. Results

### 3.1. DEGs and qRT-PCR Analysis between A. correntina and GH85

By sequencing, low-quality reads were removed, and a total of 95.23 Gb clean data was obtained (Appendix A). There were 9740 and 9489 DEGs generated from the root of *A. correntina* and GH85 on the 30th and 60th day after hydroponic culture, respectively (Figure 1a). Moreover, compared with GH85, there were fewer up-regulated DEGs in *A. correntina* (Figure 1b). Using GO function annotation analysis, DEGs of two peanut varieties were significantly enriched into GO subcategories involving biological process, cellular component and molecular function (Appendix A). Based on KEGG class analysis, DEGs of two peanut varieties were significantly enriched in the pathways of phenylpropanoid biosynthesis, carbon metabolism, purine metabolism, biosynthesis of amino acids, starch and sucrose metabolism, endocytosis, ribosome, pentose phosphate pathway, glyoxylate and dicarboxylate metabolism (Appendix A).

To verify the accuracy of RNA-seq data, we selected 10 DEGs to design primers for qRT-PCR related to the pathways of phenylalanine tyrosine and tryptophan biosynthesis, biosynthesis of amino acids, citrate cycle (TCA cycle) and phenylpropanoid biosynthesis (Appendix A). Linear regression analysis showed that the RT-PCR data were significantly correlated with the RNA-seq data (R^2^ = 0.956) (Figure 1c) (Appendix A), which indicated that the expression trend of DEGs between RNA-Seq and qRT-PCR was basically consistent. The results implied that RNA-seq data in this study were reliable.

### 3.2. DEMs Analysis between A. correntina and GH85

According to the test results of metabolites, a total of 188 root exudates with significant differences were detected in the two peanut varieties (Appendix A). Compared with *A. correntina*, the number of up-regulated DEMs in GH85 was relatively higher 0–30 days and 30–60 days after hydroponics, and the number of up-regulated DEMs of GH85 were 149and 129, respectively, in the two periods, which account for 79.3% and 68.6% of the total DEMs of two peanut varieties, respectively (Appendix A). Data revealed that there were significant differences in the content of amino acids and their derivatives, phenolic acids, organic acids, nucleotides and alkaloids, and lipids between *A. correntina* and GH85 root exudates (Appendix A). PCA analysis showed that root exudates of two peanut varieties were significantly separated at two stages (Figure 2), indicating that root exudates of the two peanut varieties were significantly different. As shown in Figure 3, stratified clustering of samples by heat map also showed that there were significantly different in root exudates of *A. correntina* and GH85 at the two stages. By KEGG enrichment analysis, DEMs of *A. correntina* and GH85 were significantly enriched in the pathways of phenylalanine metabolism, 2-Oxocarboxylic acid metabolism, pyrimidine metabolism, glucosinolate biosynthesis, biosynthesis of amino acids, ABC transporters, amino sugar and nucleotide sugar metabolism, biosynthesis of secondary metabolites and aminoacyl-tRNA biosynthesis (Appendix A).

### 3.3. Transcriptome and Metabolomics Association Analysis of A. correntina and GH85

Combining the transcriptome and metabolome information, both the DEGs and DEMs of two peanut varieties were closely related to the pathways of carbon metabolism, biosynthesis of amino acids, phenylpropanoid biosynthesis, ABC transporters, 2-oxocarboxylic acid metabolism, aminoacyl-tRNA biosynthesis, glycerophospholipid metabolism, starch and sucrose metabolism in two periods (Appendix A) (Figure 4). Moreover, significantly more DEGs and DEMs were down-regulated in these metabolic pathways of *A. correntina* compared with GH85 in two periods, except the pathways of ABC transporters, starch and sucrose metabolism, glycerophospholipid metabolism and phenylpropanoid biosynthesis at the 30th day after hydroponics (Appendix A). KEGG enriched analysis of transcriptome and metabolism showed that the DEGs and DEMs of two peanut varieties were also significantly enriched in the pathways of amino acid transport and metabolism, phenylpropanoid biosynthesis, TCA cycle, aminoacyl-tRNA biosynthesis and 2-oxocarboxylic acid metabolism (Appendix A), which closely related to biosynthesis and metabolism of amino acid and phenolic acid [25,26].

### 3.4. Interaction of Two Peanut Root Exudates and Two Pathogens

There are few studies on the interaction between peanut root exudates and pathogens. As an exploration, we applied three peanut root exudate treatments (1%, 5% and 30%) to investigate different resistances to pathogens between *A. correntina* and GH85. The results revealed that the effect of peanut root exudates on the growth of two pathogens showed concentration effects from growth promotion to inhibition. Two concentration treatments (1% and 5%) of GH85 root exudates promoted the growth of *R. solanacearum* and *F. moniliforme* compared with the control, while the same treatments of *A. correntina* root exudates showed an inhibited effect (Table 1). Concentration treatment (30%) of root exudates of two peanut varieties inhibited the growth of *R. solanacearum* and *F. moniliforme*, relative to controls, and the inhibitory effect of GH85 root exudates on *F. moniliforme* was significantly less than that of *A. correntina* (Table 1). 

### 3.5. Effects of Exogenous Phenolic Acids and Amino Acids on the Growth of R. solanacearum and F. moniliforme

The results showed that 3-(4-hydroxyphenyl)-propionic acid, syringic aldehyde, p-coumaric acid, ferulic acid, tryptophan and L-proline promoted the growth of *R. solanacearum* at 0.001 nM treatment concentration but did not reach a significant level, while the same treatments of 3-aminosalicylic acid, 2,4-dihydroxybenzoic acid, 2-methoxybenzoic acid, L-valine, L-methionine and L-aspartic acid inhibited the growth of *R. solanacearum*. The majority of treatments of phenolic acids and amino acids inhibited the growth of *R. solanacearum* under 0.1 and 0.01 nM concentration treatments except for 0.01 nM concentration treatments of L-valine and L-methionine, and there was a general trend that the inhibitory effect was stronger with the increase of concentration treatments. Among them, the inhibitory effect of 2,4-dihydroxybenzoic acid, 3-(4-hydroxyphenyl)-propionic acid, 2-methoxybenzoic acid, p-cumaric acid and ferulic acid were significantly higher than control.

The majority of treatments of phenolic acids and amino acids inhibited the growth of *F. moniliforme* under 0.1 nM concentration but showed promoting effects under 0.01 and 0.001 nM concentrations. High levels of syringic aldehyde, p-coumaric acid and ferulic acid (0.1 nM) significantly inhibited the growth of *F. moniliforme*. There was not asignificant increase in the growth of *F. moniliforme* under 0.01 and 0.001 nM concentration treatments of amino acids and phenolic acid.

## 4. Discussion

### 4.1. Root Exudates of A. correntinaInhibited the Growth of R. solanacearum and F. moniliforme

Root exudates can provide essential carbon and nitrogen sources for pathogens, which aggravate the infection of pathogens on susceptible crops under continuous cropping systems [4,27]. Previous studies show that *A. correntina* has a higher resistance to continuous cropping obstacles than cultivated varieties, closely correlating with its resistance to pathogens infection [11]. However, the results of pathogen inoculation on injured roots show that *A. correntina* does not show stronger disease resistance than cultivated varieties [11]. Therefore, we speculated that the mechanism of *A. correntina* against pathogens might attribute to its root exudates regulating soil microorganisms. For example, the fewer releases of protein and phenolic acids in root exudates restrain the growth and tropism of the rhizosphere pathogens [4]. In this study, significant differences in the composition and content of root exudates between *A. correntina* and GH85 might be the cause of their difference in resistance to pathogens.

Compared with *A. correntina*, root exudates of GH85 obviously promoted the growth of *R. solanacearum* and *F. moniliforme* under low-concentration treatments (1% and 5%) (Table 1). This indicated that *A. correntina* root exudates components might contain substances that inhibit the growth of *R. solanacearum* and *F. moniliforme* within a certain concentration range. Additionally, this good effect might be the result of the combined action of multiple substances in a certain concentration range. Results revealed that 30 percent volume of root exudates treatments of *A. correntina* and GH85 significantly inhibited the growth of two pathogens (Table 1), implying that a certain high concentration of peanut root exudates could inhibit the growth of pathogens. This might be because the high-density plant hydroponics adopted in this study far exceeded the concentration of peanut root exudates in soil under normal cultivation. In addition, this experiment was conducted in the absence of pathogen stress conditions. The infection of pathogenic microorganisms to crop roots destroyed the permeability of the root cell membrane, changed the metabolism of plants and increased the secretion released from roots to soil, which is beneficial for increasing numbers of soil pathogenic microorganisms and reducing the number of soil bacteria around the root [4]. Meanwhile, the uptake and utilization of root exudates by soil microorganisms may cause changes in soil substances in rhizosphere soil, thus affecting the structure of the soil microbial community in rhizosphere soil [28]. The effects of such changes on peanut disease resistance should be further studied.

### 4.2. Phenolic Acid and Amino Acid Might Play an Important Role in Different Effects of Root Exudates between A. correntina and GH85 on the Growth of R. solanacearum and F. moniliforme

KEGG analysis of transcriptome and metabolism showed that DEGs and DEMs of *A. correntina* and GH85 were both significantly enriched in metabolic pathways responsible for the biosynthesis and metabolism of amino acid and phenolic acid involved in the pathways of amino acid transport and metabolism, phenylpropanoid biosynthesis, TCA cycle, aminoacyl-tRNA biosynthesis and 2-oxocarboxylic acid metabolism (Appendix A) [25,26]. Further, the inhibitory effect of exogenous amino acids and phenolic acids on *R. solanacearum* and *F. moniliforme* shows concentration effects from growth promotion to inhibition as with the root exudates. The results suggested an important role of phenolic acid and amino acid in peanut defense against *R. solanacearum* and *F. moniliforme.*

### 4.3. Phenolic Acids Have a Critical Function on the Growth of Pathogens

Originating from plant root exudates or soil microbial metabolites in soil, most phenolic acids are allelochemical and can affect seed germination, plant growth, plant cell division, soil microbial structure and pathogen activity [29,30,31], including erucic acid, syringic acid, vanillin, coumaric acid, gallnut acid, p-phthalic acid, phthalic acid, caffeic acid, ferulic acid, benzoic acid, salicylic acid and cinnamic acid [32,33,34,35]. Our results showed that seven phenolic acids detected from peanut root exudates had different degrees of promoting effects on the growth of *R. solanacearum* and *F. moniliforme* (Figure 5) (Table 2). The reason might be that these substances could act as the nutrient carbon source of pathogenic microorganisms to stimulate the growth of pathogens and weaken the biological activity of some antagonistic bacteria [29,36].

Here, metabolomics results showed that the phenolic acid content and the number of up-regulated DEMs involved in the pathway of phenolic acid metabolic in root exudates of GH85 were generally higher than that of *A. correntina* (Appendix A). This might be one of the reasons for root exudates of GH85 to more effectively promote the growth of *R. solanacearum* and *F. moniliforme* than that of *A. correntina*. By transcriptomic sequencing results, more DEGs related to the metabolism of phenylpropanoid were up-regulated in GH85 compared with *A. correntina*, involved in up-regulated enzyme genes related to caffeic acid 3-o-methyltransferase, 4-coumarate: coenzyme ligase, trans-cinnamate 4-monooxygenase and ferulate -5-hydroxylase (Appendix A). The up-regulation of these enzyme genes triggers the biosynthesis and transportation of phenolic acids such as p-cumaric acid and ferulic acid [37,38,39]. This is a possible cause of the increase in the level of phenolic acid in GH85 root exudates. In addition, up-regulated DEGs related to the pathway of phenylalanine and tyrosine metabolisms in GH85 could also promote the catalytic synthesis of hydroxycinnamic acid and other phenolic acid compounds (Appendix A) [40,41].

### 4.4. Amino Acids Are Closely Related to Plant Disease Resistance

Free amino acids in plant tissues can respond directly or indirectly to biotic or abiotic stresses, including stresses of drought, cold, pruning, salt, heavy metals, temperature, air pollution, water pressure, nutrition, pests and diseases [42,43]. Most plants will adjust nitrogen metabolism by increasing or decreasing the concentration of amino acids in face of environmental pressures, reducing the damage to plants [43,44]. Total free amino acids can vary in different resistant varieties of crops, and the total amino acid content of susceptible varieties may be higher than those of resistant varieties. Relevant studies reveal that watermelon’s resistance to disease is inversely proportional to its amino acid content; the total amino acid content of susceptible watermelons is 10 times that of resistant watermelon [36]. Similarly, the amino acid content in root exudates of GH85 was higher than that of *A. correntina* (Appendix A), implying that amino acid might be closely related to GH85′s growth-promoting effect on pathogens. The reason might be that amino acids can be used as carbon sources in the low concentration range by microorganisms to promote their growth, but the process of amino acid absorption and utilization still needs further study.

Fewer up-regulations of DEGs and DEMs related to amino acid metabolism pathways were likely responsible for lower amino acid content in root exudates of *A. correntina* compared with GH85 (Appendix A, Appendix A). In addition, more amino acid synthesis also promoted the up-regulated DEGs associated with the pathway of aminoacyl-tRNA biosynthesis in GH85, including enzyme genes with amino acids as substrates, such as prolyl-tRNA synthetase, lysyl-tRNA synthetase, arginyl-tRNA synthetase, phenylalanyl-tRNA synthetase α chain, glutamyl-tRNA synthetase and isoleucyl-tRNA synthetase (Appendix A), which were conducive to the amino acid transferring to the ribosome for protein synthesis [45]. Meanwhile, the up-regulation of genes related to the pathways of tyrosine metabolism and arginine biosynthesis in GH85 (Appendix A) also promoted the up-regulation of related genes in the TCA cycle pathway, including the promotion synthesis of succinic acid and fumaric acid (Appendix A) [25,26].

In this study, the interaction of five DEMs (Tryptophan, L-Proline, L-Valine, L-Methionine, L-Aspartic Acid) related to amino acids (Figure 6) with *R. solanacearum* and *F. moniliforme* showed concentration effects from growth promotion to inhibition in agreement with the results of root exudates. It was further suggested that amino acids might have a vital role on the effect of peanut root exudates on the growth of two pathogens. Relevant studies have shown that glutamic acid, phenylalanine, glycine, aspartic acid and alanine can significantly promote the growth of cucumber fusarium wilt [46], consistent with the results of five amino acid interactions with two pathogens in this study (Table 2).

## 5. Conclusions

*A. correntina* root exudates inhibited the growth of *R. solanacearum* and *F. moniliforme* more effectively than GH85, which might be because there were fewer up-regulated DEGs and DEMs in *A. correntina* compared with GH85. DEGs and DEMs between *A. correntina* and GH85 were both enriched in the pathways of amino acid and phenolic acid metabolism. The effects of exogenous amino acids and phenolic acids on *R. solanacearum* and *F. moniliforme* were the same as root exudates, which showed a concentration effect from promoting growth to inhibiting growth. The result suggested that amino acids and phenolic acids might play an important role in *A. correntina*’s inhibition of the growth of two pathogens.

## Figures and Tables

**Figure 1 genes-14-00528-f001:**
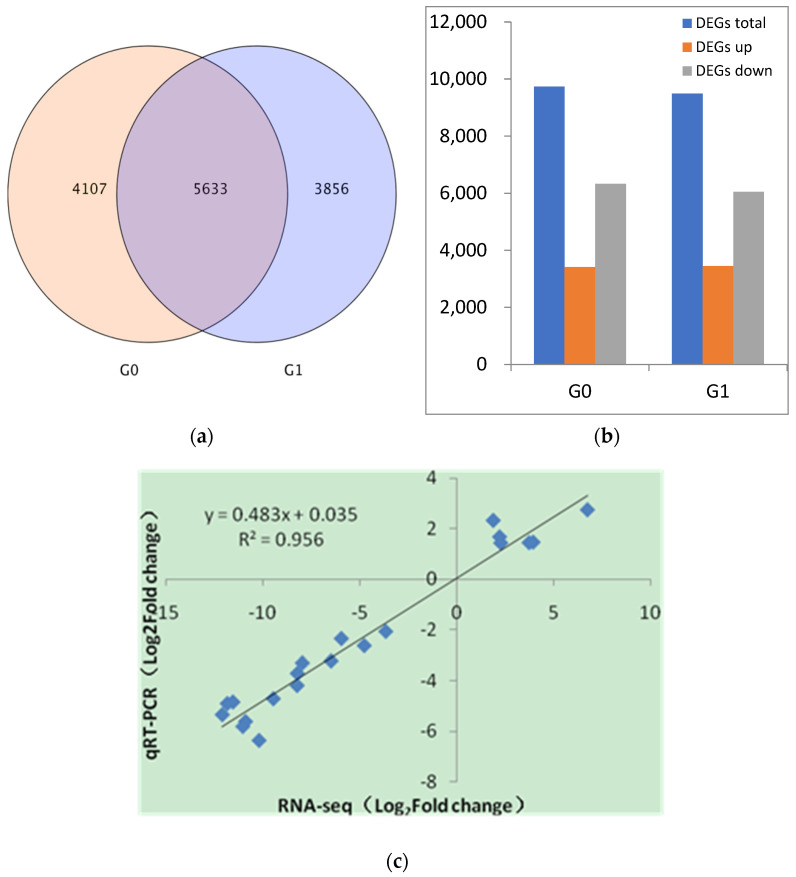
The number of DEGs between *A. correntina* and GH85 and qRT-PCR validation of selected DEGs. (**a**) Venn diagram of DEGs. G0: *A. correntina*-30d vs. GH85−30d. G1: *A. correntina*−60d vs. GH85−60d. (**b**) The number of DEGs between *A. correntina* and GH85. (**c**) Linear regression analysis of RT-PCR and RNA-Seq data.

**Figure 2 genes-14-00528-f002:**
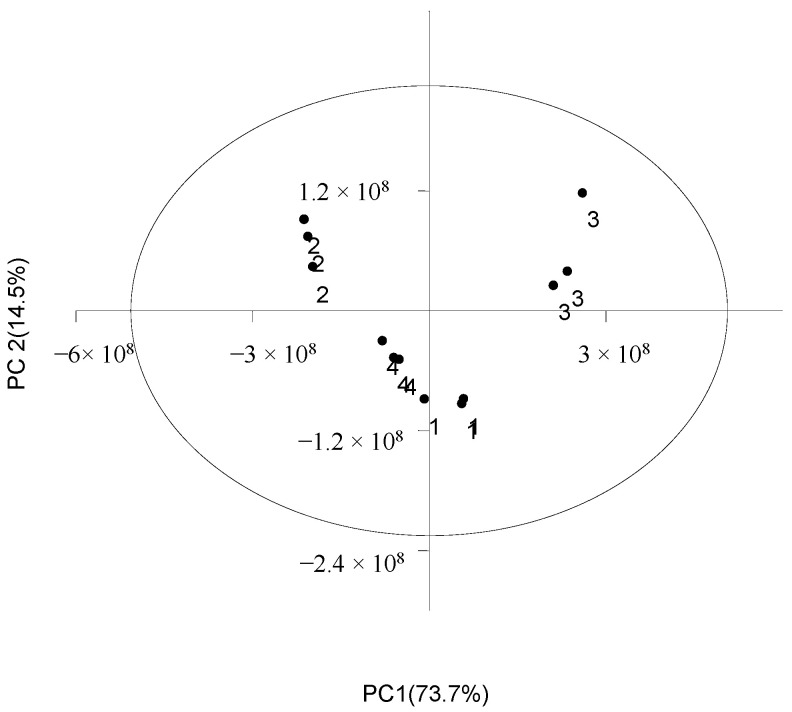
PCA analysis of root exudates of two peanut varieties. 1: *A. correntina*−30d. 2: *A. correntina*−60d. 3: GH85−30d. 4: GH85−60d.

**Figure 3 genes-14-00528-f003:**
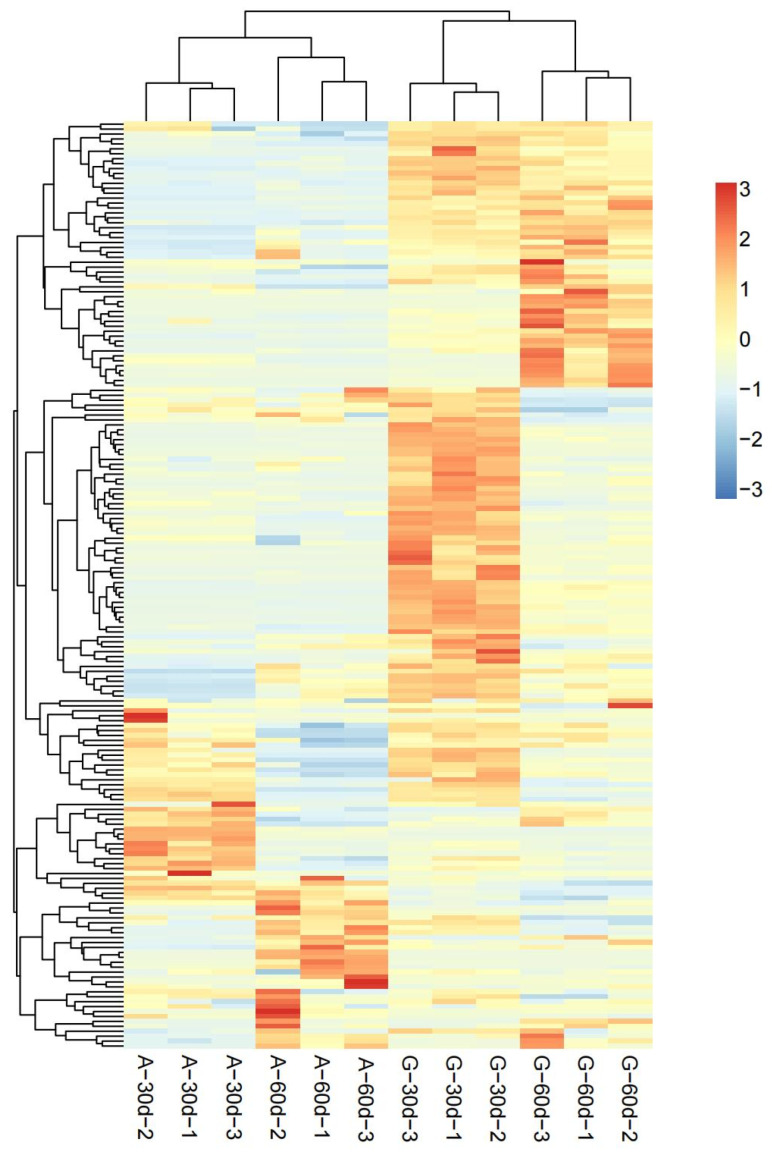
Clustering analysis diagram of DEMs between *A. correntina* and GH85. A−30d: *A. correntina*−30d; A−60d: *A. correntina*−60d; G−30d: GH85−30d; G−60d: GH85−60d (the same below).

**Figure 4 genes-14-00528-f004:**
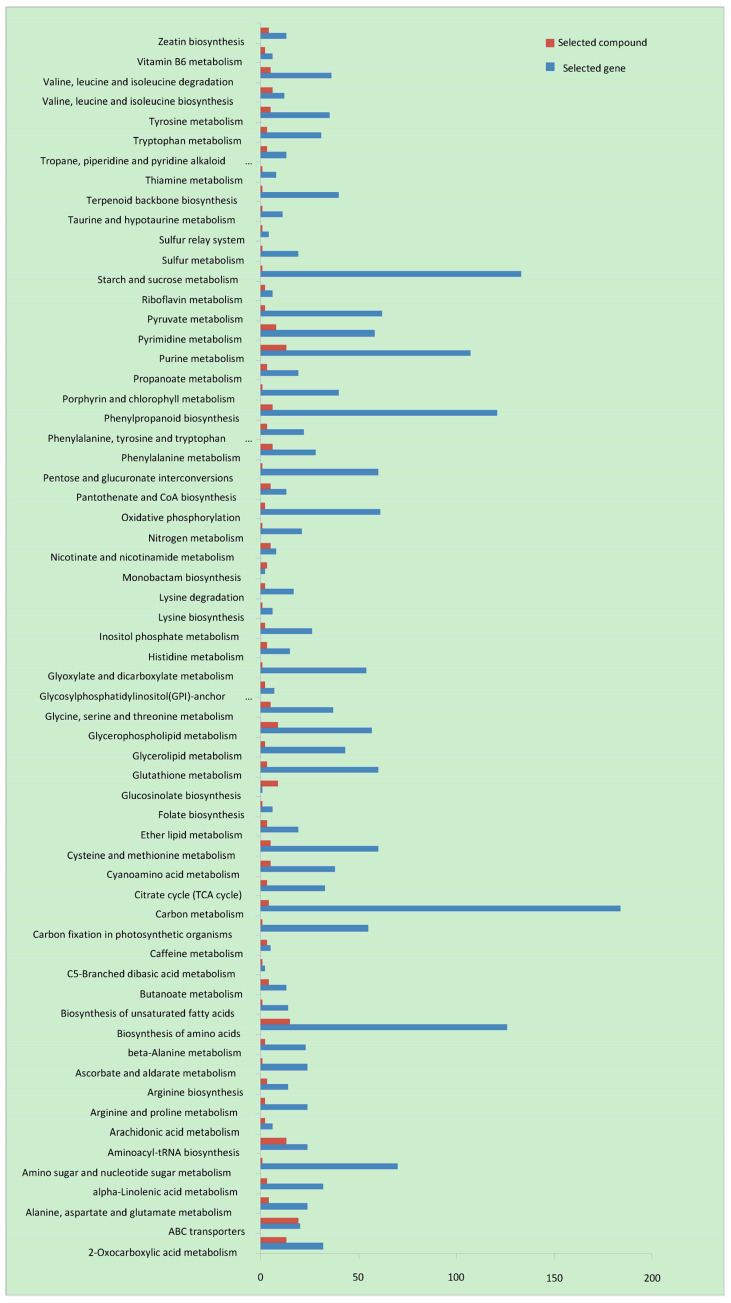
KEEG classification of DEGs associated with DEMs between *A. correntina* and GH85.

**Figure 5 genes-14-00528-f005:**
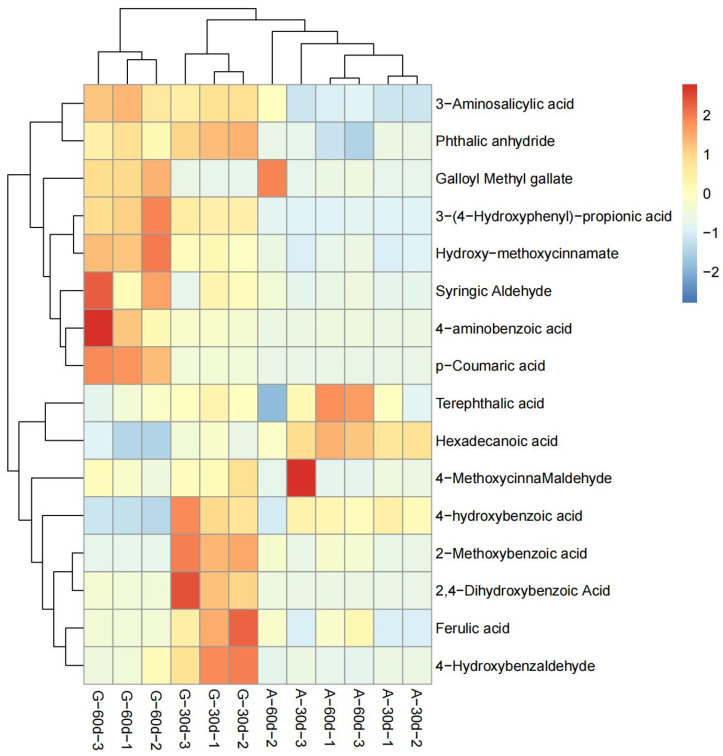
Clustering analysis diagram of DEMs related to phenolic acids between *A. correntina* and GH85.

**Figure 6 genes-14-00528-f006:**
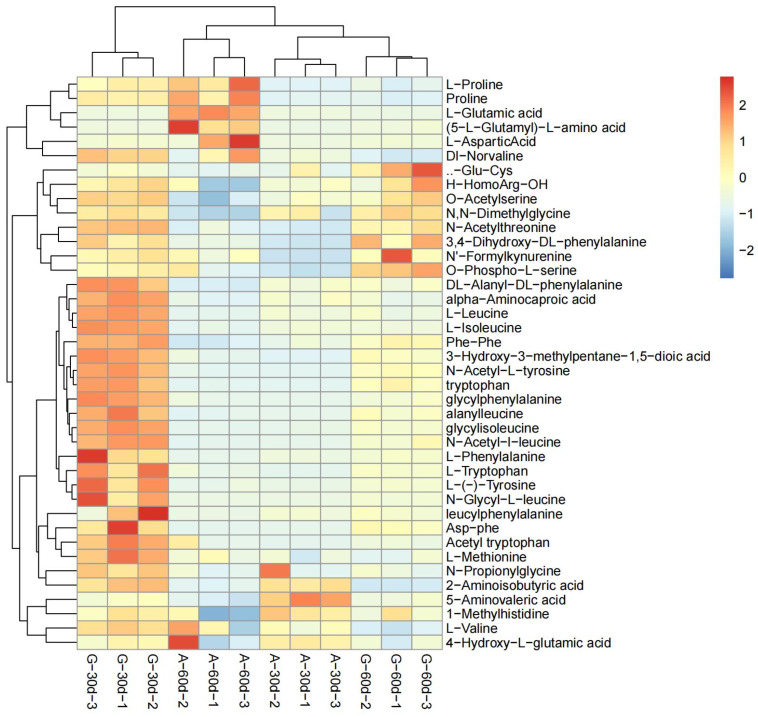
Clustering analysis diagram of DEMs related to amino acid between *A. correntina* and GH85.

**Table 1 genes-14-00528-t001:** Root exudates of peanuts interaction with *R. solanacearum* and *F. moniliforme*.

		*F. moniliforme*				*R. solanacearum*	
Treatment	Concentration (%)	3d (cm)	Inhibition (%)	4d (cm)	Inhibition (%)	1d (10^6^ cfu/mL)	Inhibition (%)
Control	1	4.17 ± 0.06	-	5.20 ± 0.10	-	8.79 ± 0.20	-
	5	4.03 ± 0.06	-	5.03 ± 0.06	-	8.58 ± 0.32	-
	30	4.00 ± 0.10	-	4.93 ± 0.15	-	7.87 ± 0.39	-
GH85 30d	1	4.37 ± 0.06	−4.8	5.40 ± 0.10	−3.85	9.31 ± 0.36	−5.99
	5	4.50 ± 0.10	−11.66	5.33 ± 0.06	−5.96	9.08 ± 0.24	−5.75
	30	3.70 ± 0.10	7.5	4.50 ± 0.20	8.72	6.50 ± 0.30	17.4
GH85 30–60d	1	4.50 ± 0.10	−7.91	5.50 ± 0.10	−5.77	9.36 ± 0.30	−6.49
	5	4.40 ± 0.10	−9.18	5.30 ± 0.10	−5.37	8.68 ± 0.23	−1.09
	30	3.60 ± 0.10	10.00	4.57 ± 0.15	7.3	6.10 ± 0.26 *	22.49
*A. correntina* 30d	1	4.07 ± 0.06	2.4	5.07 ± 0.06	2.5	8.51 ± 0.39	3.11
	5	3.77 ± 0.12	6.45	4.73 ± 0.06	5.96	8.01 ± 0.05	6.72
	30	3.37 ± 0.12	15.75	4.00 ± 0.10 *	18.86	7.51 ± 0.44	4.57
*A. correntina* 30–60d	1	4.17 ± 0.06	0.00	5.00 ± 0.10	3.85	8.48 ± 0.24	3.53
	5	3.8 ± 0.10	5.70	4.83 ± 0.15	3.98	8.07 ± 0.21	5.98
	30	3.23 ± 0.06 *	19.25	4.13 ± 0.06	16.23	7.39 ± 0.12	6.14

Note: Superscript * mean significant difference between treatment and control (*p* < 0.05). The same below.

**Table 2 genes-14-00528-t002:** Effects of exogenous phenolic acids and amino acids on growth of *R. solanacearum* and *F. moniliforme*.

		*F. moniliforme*				*R. solanacearum*	
Treatment	Concentration (nM)	3d (cm)	Inhibition Rate (%)	4d (cm)	Inhibition Rate (%)	1d (10^6^ cfu/mL)	Inhibition Rate (%)
Control		3.47 ± 0.12		4.73 ± 0.06		2.90 ±0.06	
3-Aminosalicylic Acid	0.001	3.90 ± 0.20	−12.39	4.80 ± 0.20	−1.48	2.86 ± 0.09	1.72
	0.01	3.43 ± 0.12	1.15	4.87 ± 0.12	−2.96	1.88 ± 0.16 *	35.17
	0.1	3.21 ± 0.10	7.49	4.60 ± 0.10	2.75	1.34 ± 0.30 *	53.79
2,4-Dihydroxybenzoic Acid	0.001	3.67 ± 0.06	−5.76	4.90 ± 0.10	−3.89	2.44 ± 0.21	15.86
	0.01	3.77 ± 0.06	−8.65	5.17 ± 0.06	−9.3	0.38 ± 0.18 *	87.24
	0.1	3.33 ± 0.06	4.03	4.37 ± 0.12	7.61	0.00 ± 0.00	-
3-(4-Hydroxyphenyl)-propionic Acid	0.001	3.67 ± 0.06	−5.76	4.97 ± 0.15	−5.07	3.00 ± 0.15	−3.45
	0.01	3.80 ± 0.10	−9.51	5.13 ± 0.15	−8.46	1.56 ± 19.52 *	46.21
	0.1	3.57 ± 0.06	−2.88	4.87 ± 0.06	−2.96	0.00 ± 0.00	-
Syringic Aldehyde	0.001	3.67 ± 0.06	−5.76	4.90 ± 0.10	−3.59	2.95 ± 0.08	−1.72
	0.01	3.33 ± 0.06	4.03	4.80 ± 0.10	−1.48	1.60 ± 0.04 *	44.94
	0.1	2.63 ± 0.06	24.21	3.33 ± 0.06 *	29.6	1.18 ± 0.14 *	59.31
2-Methoxybenzoic acid	0.001	3.63 ± 0.06	−4.61	4.87 ± 0.06	−2.96	1.85 ± 0.11 *	36.21
	0.01	3.33 ± 0.06	4.03	4.60 ± 0.10	2.75	46.67 ± 0.09 *	83.91
	0.1	3.43 ± 0.15	1.15	4.70 ± 0.10	0.63	0.00 ± 0.00	-
p-Coumaric Acid	0.001	3.80 ± 0.10	−9.51	5.10 ± 0.10	−7.82	3.13 ± 0.10	−7.93
	0.01	3.13 ± 0.21	9.8	3.83 ± 0.06	19.03	1.22 ± 0.14 *	57.82
	0.1	0.50 ± 0.00	-	0.50 ± 0.00	-	0.000 ± 0.00	-
Ferulic Acid	0.001	3.83 ± 0.06	−10.37	5.13 ± 0.06	−8.46	3.04 ± 0.12	−4.71
	0.01	3.37 ± 0.12	2.88	4.30 ± 0.17	9.09	1.77 ± 0.12 *	39.08
	0.1	0.50 ± 0.00	-	0.50 ± 0.00	-	0.00 ± 0.00	-
Tryptophan	0.001	3.70 ± 0.00	−6.62	4.83 ± 0.06	−2.11	3.00 ± 0.12	−3.45
	0.01	3.87 ± 0.06	−11.53	5.13 ± 0.06	−8.46	2.13 ± 0.18	26.78
	0.1	3.47 ± 0.06	0	4.67 ± 0.06	1.28	1.16 ± 0.13 *	60.11
L-Proline	0.001	3.47 ± 0.06	0	4.90 ± 0.02	−3.59	2.97 ± 0.20	−2.52
	0.01	3.77 ± 0.06	−8.65	4.93 ± 0.06	−4.23	2.16 ± 0.21	25.52
	0.1	3.70 ± 0.00	−6.63	5.07 ± 0.06	−7.19	2.72 ± 0.14	6.21
L-Valine	0.001	3.50 ± 0.10	−0.86	4.77 ± 0.15	−0.85	2.68 ± 0.14	7.59
	0.01	3.93 ± 0.12	−13.26	5.33 ± 0.06	−12.68	3.04 ± 0.20	−4.83
	0.1	3.67 ± 0.12	−5.76	5.03 ± 0.06	−6.34	1.39 ± 0.16 *	52.07
L-Methionine	0.001	3.60 ± 0.10	−3.75	4.77 ± 0.06	−0.85	2.87 ± 0.06	0.92
	0.01	3.50 ± 0.10	−0.86	4.50 ± 0.10	4.86	3.26 ± 0.0 8	−12.3
	0.1	3.30 ± 0.00	4.9	4.03 ± 0.15	14.8	2.19 ± 0.21	24.6
L-Aspartic Acid	0.001	3.60 ± 0.10	−3.75	4.80 ± 0.10	−1.48	2.79 ± 0.08	3.68
	0.01	3.53 ± 0.06	−0.73	4.77 ± 0.06	−0.85	1.24 ± 0.13 *	57.36
	0.1	3.23 ± 0.06	6.92	4.60 ± 0.10	2.75	0.57 ± 0.14 *	80.46

## Data Availability

The authors affirm that all data necessary for confirming the conclusions of the article are present within the article, figures, and tables.

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
