# Peer review of "Root Metabolism and Effects of Root Exudates on the Growth of Ralstonia solanacearum and Fusarium moniliforme Were Significantly Different between the Two Genotypes of Peanuts"

_genes, 2023, doi:10.3390/genes14020528_

Round 1
Reviewer 1 Report
[genes-2117814-review]
Comments and Suggestions for Authors:
This paper is of great significance to reveal the continuous cropping resistance mechanism of A. correntina, and it also have instructive significance on genetic improvement of peanut cultivars against continuous cropping obstacle.
Here are some suggestions:
Ø How did the authors confirm root exudates concentration of 1%, 5% and 30%? In my view, the difference between 5% and 30% is huge. The authors may explain in methods or discussion.
Ø How about the concentration of 12 compounds in the root exudates? Their treatment effect was similar, I wonder their concentration difference between root exudates and exogenous treatment.
Ø The 12 compounds should be described in the methods.
Ø Figures in the paper are suggested to give readers more useful information. Besides, the experimental pictures are also very important. If the authors keeps any, it is suggested to show some helpful test pictures.
Ø The source of the reference genome should be clearly marked.
Ø The main-text can be more concise. Some sentences have repetitive descriptions, especially in the section materials and methods.
Ø “DGEs” or “DEGs”? Both of them appear in the text. I think “DEGs” is more appropriate.
Ø A few spelling mistakes, such as "ranscriptome" (line28), please check the full text carefully.
Author Response
Thank you very much for your revision and suggestions on our paper.
In view of the above questions, we have made article by article modifications.
- Related descriptions of root exudates and 12 compounds were added.
2, The paper adds the source of peanut reference genome.
- The full paper has been modified as suggested.

Reviewer 2 Report
The work is interesting but the paper requires extensive editing- please seek a proficient scientific edit. The writing prevents clear understanding of the work
What is strong in this research is the use of the authentic chemicals in showing inhibition of the pathogens' growth - this could be a starting point. Mixes here should be tried out
What detracts though is the lack of any normalization of the root exudates obtained through hydroponic growth. IF the wild type grew less or had less root hairs or there were differential microbial constituents then this could explain the results
There were no supplemental data although they were referenced.
methods on how plants were grown etc inadequate
you need proof of the sterility of the hydroponic growth

Author Response
Thank you very much for your revision and suggestions on our paper.
The full paper has been modified as suggested. This study is the starting point of studying the resistance mechanism of wild peanut species A. correntina. Based on this, further research will be carried out. we did not do the morphological determination of peanut roots ,Because we don't want to destroy peanut roots and affect their root secretion activities. It is difficult to maintain sterility of hydroponic growth in this study. The hydroponic solution without peanut cultivation was set as the control, and no substance was detected in the control. In 2022, tests in root-rot infested soil showed that A. correntina also showed stronger resistance than peanut cultivar, found differences in their metabolites in rhizosphere soil and root transcriptome, Relevant data is being collated for publication. It further indicated the disease resistance of A. correntina. For all this , there are still many research contents needed to be clarified, as you suggest.

Reviewer 3 Report
1. Why have the authors chosen particular peanut genotypes. Indicate the literature reference for its importance and features.
2. The sampling for RNA seq and qRT PCR is not clear. Clearly mention that how samples were prepared.
3. Figures 1 and 4 representation and also resolution need to improve
4. The manuscript suffers from a long introduction and discussion, the first paragraph of the introduction should be precise, and the final paragraph of the introduction should clearly explain the scope, importance, and incentive of the work in a manner that is understandable to the objectives
5. In discussion, point out the major gens with previous studies. The final paragraph of the discussion should clearly describe the main conclusions of the work, their importance, and potential for further studies.
6. I have found numerous English language problems through the manuscript text, better polish the language.
Author Response
Thank you very much for your revision and suggestions on our paper.
In view of the above questions, we have made article by article modifications.
- We added explanations of two peanut genotypes.
2, we added the sampling method of RNA sequence and qRT PCR
- Figure 1 and Figure 4 were made in Microsoft Excel. If the resolution is not up to scratch, we may consider putting them in the supplementary fig.
- The full paper has been modified as suggested.

Round 2
Reviewer 2 Report
The revision still suffers from serious problems with english presentation.
But scientifically it is unsound because no where is there any normalization for the metabolites in the root exudates Cannot accept without this vital information.
Also in the growth studies with the pathogens, the root exudates were amended with other media ; this is not a valid method.
Author Response
Thank you very much for your guidance on our article. We have revised the article for English language problems. In metabolome detection, the experimental data were corrected by setting blank and deducting control impurities, so as to ensure the accuracy of the data as far as possible. The purpose of expanding microorganism culture could not be achieved by solely relying on root exudates culture. Therefore, mixed culture of root exudates and medium was adopted to reveal the role of root exudates through comparison with control. This study can provide reference and ideas for further detailed and in-depth study.

Reviewer 3 Report
Accept in present form
Author Response
Thank you very much for your guidance on our article. We have revised the article for English language problems.
